# Efficiency Evaluation and Influencing Factors of Green Innovation in Chinese Resource-Based Cities: Based on SBM-Undesirable and Spatial Durbin Model

**DOI:** 10.3390/ijerph192113772

**Published:** 2022-10-23

**Authors:** Yaguai Yu, Zanzan Xu, Panyi Shen, Lening Zhang, Taohan Ni

**Affiliations:** 1Business School, Ningbo University, Ningbo 315211, China; 2Donghai Academy, Ningbo University, Ningbo 315211, China; 3Business School, University of Nottingham, Ningbo 315199, China

**Keywords:** green innovation in resource-based cities, efficiency evaluation, influencing factors, super-efficiency SBM model with undesired outputs, spatial Durbin model

## Abstract

Based on data from 64 resource-based cities in China from 2010 to 2019, the efficiency of green innovation is evaluated by using the super-efficiency SBM Model with undesired outputs, while influencing factors of green innovation efficiency are analyzed by the spatial Durbin model. The results are as follows. First, as for the efficiency evaluation, the average green innovation efficiency in 62 resource-based cities from 2010 to 2019 is only 0.5689, while the green innovation efficiency of declining cities is the highest, and the growth type is the lowest in the comprehensive planning cities. Second, based on spatial self-correlation in resource-based cities, the government support, and the influencing factors including the industrial structure and economic development, have positive impacts, while the environmental regulations and opening to the outside world will inhibit the urban green innovation. Therefore, to enhance the green innovation efficiency in resource-based cities, some suggestions include formulating differentiated development strategies, forming regional cooperation mechanisms, increasing government scientific and technological support, determining the reasonable intensity of environmental regulations, setting entry barriers for polluting enterprises, and optimizing industrial structure.

## 1. Introduction

Green innovation is at the core of the transformation of traditional cities to green cities, and the green innovation efficiency is an important performance for cities to maintain a sustainable economic competitive advantage. Green innovation in cities is essentially an action to transform the economic development of cities from the traditional mode to the combined mode of environmental friendliness and the efficient use of resources. With green friendliness as the start of the path and scientific and technological innovation as the driving force and key point, the green innovation efficiency is the ultimate manifestation of this action. Resource-based cities mainly refer to a special type of city that rises from resource extraction and relies mainly on resource-based industries to support their economic development [1]. In 2013, the State Council promulgated the National Sustainable Development Plan for Resource-based Cities (2013–2020), which identified 262 resource-based cities, including 116 prefecture-level cities, accounting for 39.59% of the total number of prefecture-level cities in China. With the continuous exploitation of resources, resource-based cities have gradually formed a single industrial structure that relies too much on non-renewable resources and natural resources, leading to a slow economic growth, an ecological environment deterioration, and a serious loss of highly qualified talents [2]. The traditional economic development approach will be difficult to continue, and it is worth exploring how resource-based cities can achieve sustainable development.

Enhancing the green innovation capacity of resource-based cities can help resolve the multiple crises which are being faced by the development of resource-based cities and promote their sustainable development [3]. To achieve the improvement of the green innovation capacity of resource-based cities, it is necessary to build based on the objective evaluation and analysis of the existing green innovation capacity of resource-based cities and to analyze the factors affecting the size of the green in-novation efficiency of resource-based cities based on the existing level of green innovation before an effective improvement plan can be formulated. Therefore, on the one hand, it is necessary to reasonably evaluate the green innovation efficiency of resource-based cities to grasp the actual level of green innovation in China’s resource-based cities and its specific differences among various types of resource-based cities; on the other hand, it is necessary to explore the relevant factors affecting the green innovation efficiency of resource-based cities to provide a scientific cracking path for the problem of improving the green innovation level of resource-based cities.

This paper evaluates the efficiency and influencing factors of green innovation in resource-based cities using the super-efficiency SBM model with a non-expected output, Moran’s I index, and the spatial econometric model, respectively, with the data of 64 re-source-based cities in China from 2010 to 2019 as the samples. The research in this paper finds that (1) the mean value of the green innovation efficiency of resource-based cities is low between 2010 and 2019, which is only 0.5689, among which the green innovation efficiency of declining cities is the highest and the lowest of growing cities among the integrated types of planning. (2) The green innovation efficiency of resource-based cities is spatially autocorrelated, and government support, industrial structure upgrading, and economic development will promote the green innovation efficiency of resource-based cities, while environmental regulation and opening to the outside world will inhibit the development of urban green innovation.

The contribution of this paper is that (1) current resource-based city-related studies focus on directly measuring the transformation efficiency of resource-based cities while lacking the exploration of the drivers of transformation, i.e., the green innovation efficiency [4,5], thus this paper incorporates the green innovation efficiency into the efficiency measurement system of resource-based cities and analyzes the existing level of the green innovation efficiency of resource-based cities in China and its influencing factors, which provides a basis for improving the green innovation capacity of resource-based cities and enriches the research content of resource-based cities. (2) In the current green innovation-related literature, the evaluation objects are mostly at the enterprise and industry levels and less at the prefecture-level city level. Academics have initially established indicators to measure the green innovation index at the micro level, but there is a lack of reasonable and effective measures for the green innovation level at the regional level [6,7,8]. This paper establishes the evaluation object as the prefecture-level city level of resource-based cities, focusing on the evaluation of the green innovation efficiency measurement at the regional level and expanding the research area of the green innovation efficiency. (3) Many scholars often do not take both being green and innovative into account when selecting the green innovation efficiency evaluation indicators and their influencing factors [9,10], and even though some studies are more comprehensive, they do not reflect the characteristics of the green innovation development in resource-based cities. In this paper, we will combine the construction principles of the evaluation index system and the characteristics of the green innovation efficiency of resource-based cities to build a reasonable and scientific framework of the evaluation index system and the influencing factors of the green innovation efficiency of resource-based cities from a comprehensive perspective.

The rest of this paper is organized as follows: the second part is a literature review, the third part is a research design of the green innovation in resource-based cities, the fourth part is an evaluation of the efficiency of the green innovation in resource-based cities, the fifth part is an analysis of the influencing factors of the green innovation efficiency in resource-based cities, and finally, we present the conclusions and recommendations.

## 2. Literature Review

The definition of green technology, first proposed by Ernest in 1994, is a collective term for the goods, technologies, and processes that eliminate or reduce pollution while saving resources [11]. Green innovation, on the other hand, is a concept developed based on and closely related to green technology innovation, which refers to hardware and soft technology innovation related to green products or green processes [12]. Garcia-Granero (2018) argues that the green innovation is divided into four types of innovation: the green product, green process, green organization, and green marketing innovation and should also include a complete portfolio of performance indicators under each type to measure green innovation comprehensively [13]. The meaning of the green innovation efficiency in this paper refers to the ratio of the innovation resource’s input to its output while ensuring energy consumption savings and environmental pollution control [3].

In terms of a green innovation efficiency evaluation, most of the existing studies on a green innovation efficiency evaluation follow the traditional stochastic frontier analysis method and data envelopment analysis method. For example, Yixin Zhang and Xiumei Lin (2015) combined the DEA model with the Malmquist index method to measure the inter-provincial green innovation efficiency in China from both the static and dynamic aspects [14]. Based on the DEA-Ram model, Ren and Yao (2014) made a practical measurement of the green innovation efficiency in the Shanxi Province from three systems: green, innovation, and the economy [15]. In measuring the green innovation efficiency of eight economic zones in China, Feng (2013) found that the traditional DEA model did not take into account the “slack” problem, which caused errors in the efficiency results, so he chose the DEA-SBM model as the research model and compared it with the traditional model, and found that the SBM model was more effective [16]. With the development of the continuous use of the SBM model, the advantages of the super-efficient SBM model are also more prominent, which further refines the differences among the effective units based on the SBM model and solves the problem that the traditional SBM model cannot evaluate the arrangement of the effects units. For example, Fei-Fei Zhang (2020) analyzed and measured the static efficiency and dynamic efficiency of green innovation in China’s manufacturing industry based on the combined model of the SBM model and ML index [17]. Ma Zhichao et al. (2022) also used a super-efficient SBM and ML model to measure and evaluate the green economic efficiency of the Lanzhou-Xining urban cluster [18]. The generation of data envelope models has provided a variety of research methods for the evaluation of the green innovation efficiency, and they have been continuously optimized in development.

Analyzed in terms of the relevant aspects of the green innovation efficiency, SMEs are a typical indicator that take into account both the firm and environmental characteristics and are therefore subject to the influence of multiple factors. It is generally determined that the economic level can lead to more new factors, which in turn promotes an efficient technological innovation in SMEs [19]. An environmental regulation can lead to a “pollution paradise” [20], which also pushes SMEs to develop pollution control technologies and equipment, stimulating environmental technological innovation [21], according to which it seems that the effect of the environmental regulation presents diverse characteristics. Some scholars point out that foreign funds are mostly concentrated in high environmental pollution and high energy fields, which have more negative impediments than positive advances on the effect of environmental technological innovation [22]. In addition, some researchers have considered the effect on the effectiveness of corporate environmental protection from the perspective of foreign trade and the industrial modernization of Chinese enterprises (Li, 2017). Moreover, some researchers have considered the effect on the effectiveness of the corporate environmental technology innovation efficacy from the perspective of foreign trade and the degree of the industrial modernization of Chinese enterprises [23,24], but the results vary because of the difference in scope and methodology.

In summary, domestic and foreign scholars have conducted a certain amount of research around the issues of the green innovation efficiency evaluation and the green innovation efficiency impact factors and have achieved results of a great significance which provide a rich theoretical foundation for the research of this paper, but it is not difficult to find that there is less research on the green innovation efficiency with resource-based cities as the evaluation object, and resource-based cities are facing a critical period of transformation development driven by green innovation. Although the existing literature is relatively comprehensive in terms of the index system of the green innovation efficiency evaluation, it does not reflect the characteristics of the innovation development of resource-based cities and is not suitable for the measurement and evaluation of the green innovation efficiency of resource-based cities directly. Based on this, this paper measures and analyzes the green innovation efficiency of resource-based cities and its influencing factors based on the efficiency perspective, using the relevant data of 64 prefecture-level resource-based cities from 2010–2019, and applying the super-efficiency SBM model considering the non-expected output, to provide a scientific basis for decision making to improve the green innovation efficiency of Chinese resource-based cities and realize the sustainable development of Chinese resource-based cities.

## 3. Research Design for Green Innovation in Resource-Based Cities

### 3.1. City Identification and Data Sources

#### 3.1.1. Identification of Resource-Based Cities

According to the concept in the National Sustainable Development Plan for Resource-based Cities (2013–2020) (hereinafter referred to as the Plan) issued by the General Office of the State Council in 2013, there are currently a total of 262 resource-based cities in China, of which 116 are prefecture-level cities. Given that the prefecture-level cities in the resource category often possess the characteristics of long development history and large scale, their requirements for sustainable urban development are particularly urgent. Therefore, in this paper, 64 prefecture-level cities across China are selected as the main research subjects according to the availability and completeness of research data. Meanwhile, according to the content in the Plan, these 64 resource-based cities are divided into four categories: Growth type, Mature type, Decline type, and Regeneration type. The details are shown in Table 1.

#### 3.1.2. Data Sources

Based on the scope and availability of the data, this paper selected information from 64 resource-based cities during the period from 2010 to 2019 as the database for the study. Most of the data were obtained from the China City Statistical Yearbook, EPS database, and the websites of the Department of Science and Technology of each city’s statistical yearbook. Among them, some incomplete data were compensated by the linear interpolation method.

### 3.2. Research Methodology

#### 3.2.1. Super-Efficient SBM Model Considering Non-Desired Outputs and Innovation Efficiency Evaluation

In this paper, the green innovation efficiency of resource-based cities is assessed and analyzed by an over-efficient SBM model that considers non-desired outputs. Compared with the traditional DEA model, the SBM model analyzes the slack changes in the objective function and deals with the slack relationship with input–output variables, and also examines the effect of non-desired outputs on efficiency. However, if the effective value of all the efficient decision units measured by the SBM model is one, it will lead to unrecognized value differences within all the efficient decision units, resulting in an error in the final decision. The super-efficient SBM model, on the other hand, can reclassify all the effective units with an efficiency value of one, so that the comparison of all the effective decision units can be performed, enhancing the accuracy of the decision analysis. The formula of the super-efficient SBM model considering the non-desired outputs is as follows.
(1)$$\rho =\min\frac{1}{m}\sum_{i=1}^{m}\frac{\frac{\bar{x}}{x_{\mathrm{ik}}}}{\frac{1}{r_{1}+r_{2}}}\left(\sum_{s=1}^{r_{1}}\frac{y^{d}}{y_{\mathrm{sk}}^{d}}+\sum_{q=1}^{r_{2}}\frac{\bar{y^{u}}}{y_{\mathrm{qk}}^{u}}\right)$$
(2)$$\{\begin{array}{c} \begin{array}{c} \bar{x}\geq \overset{n}{\underset{j=1,j\neq k}{\sum }}x_{\mathrm{ij}}\lambda_{j};\bar{y^{d}}\geq \overset{n}{\underset{j=1,j\neq k}{\sum }}y_{\mathrm{sj}}^{d}\lambda_{j}\\ \bar{y^{d}}\geq \overset{n}{\underset{j=1,j\neq k}{\sum }}y_{\mathrm{qj}}^{d}\lambda_{j};\bar{x}\geq x_{k};\bar{y^{d}}\geq y_{k}^{d};\bar{y^{u}}\geq y_{k}^{u} \end{array}\\ \begin{array}{c} \lambda \geq 0;\;i=1,2,\cdots ,m;j=1,2,\cdots ,n;\\ s=1,2,\cdots ,r_{1};q=1,2,\cdots ,r_{2} \end{array} \end{array}$$

In Equations (1) and (2), ρ represents the green innovation efficiency of resource-based cities; n denotes the number of resource-based cities; m is the number of inputs; $r_{1}$ and $r_{2}$ denote the desired output and non-desired output, respectively; and x, $y^{d}$, and $y^{u}$ are the elements of their corresponding input matrix, desired output matrix, and non-desired output matrix.

#### 3.2.2. Global Moran’s I Index and Spatial Divergence Treatment

In this paper, we used the global Moran’s I index analysis to test the spatial autocorrelation of the green innovation efficiency in resource-based cities. The global Moran’s I index will help to verify the spatial development pattern within the study area, expressing the general nature of the variation in the mean value of a function over the study time horizon. Its expression is:(3)$$I=\frac{n\sum_{j}^{n}\sum_{j}^{n}\left(x_{i}-\bar{x}\right)\left(x_{j}-\bar{x}\right)}{\sum_{j=1}^{n}{\left(x_{i}-\bar{x}\right)}^{2}\sum_{j}^{n}\sum_{j}^{n}\omega_{\mathrm{ij}}}$$

In Equation (3): I denotes the Moran index and${\;x}_{i}$ and $x_{j}$ denote the green innovation efficiency values of resource city i and resource city j, respectively. $\bar{x}$ is the arithmetic mean of the green innovation efficiency of all cities; n is the number of cities in the study sample; and $\omega_{\mathrm{ij}}$ is an adjacency weight matrix, which indicates the adjacency relationship between two cities.

#### 3.2.3. Spatial Econometric Model and Influence Factor Analysis

In contrast to the traditional regression methods, spatial measures take into account the complex spatial correlation and spatial dependence of the sample (Huizhong Dong, 2021) [21]. The new economic geography theory stated that the activity of technological innovation has a high aggregation effect within the urban territorial space, firstly because of the existence of industrial linkages, technological spillovers, and incremental spatial rewards within the region, and secondly because of the diffusion effect (Juchun Lu, 2021) [20], thus it is necessary to use a spatial econometric model based on geography to analyze the factors of the green innovation effect in the sample cities. The spatial Durbin model (SDM) examines not only the negative effects of the lagged causes of the dependent variable on the explanatory variables but also the negative effects of different causes on the spatial spillover effects of the explanatory variables. Therefore, we will apply the spatial Durbin model (SDM) to assess the role of the neighborhood dependent variables and independent variables on the dependent variables. The modeling structure is as described in Equation (4).
(4)$$y_{\mathrm{it}}=a_{\mathrm{it}}+\beta x_{\mathrm{it}}+\rho \sum_{j=1}^{n}\omega_{\mathrm{ij}}y_{\mathrm{jt}}+\varphi \sum_{j=1}^{n}w_{\mathrm{ij}}x_{\mathrm{jt}}+u_{i}+v_{t}+\varepsilon_{\mathrm{it}}$$

In Equation (4), the $y_{\mathrm{it}}$ denotes the green innovation efficiency of the resource-based city i in year t; $x_{\mathrm{it}}$ denotes the observed values of the main factors of the resource-based city i in year t; β is the vector of the parameters to be estimated for each factor; ρ is the spatial autoregressive coefficient, which denotes the influencing factors of the green innovation efficiency in the neighborhood; and φ denotes the weighting factor of the neighborhood factors on the green innovation effectiveness of the region. $u_{i}$ and $v_{t}$ represent the individual effects and time effects in turn, and $\varepsilon_{\mathrm{it}}$ is the error term. When φ = 0, which indicates that the green innovation efficiency of the region depends only on the neighborhood, then it can be designed as a spatial lag model (SAR). When φ = 0, there is ρ =0, the $\varepsilon_{\mathrm{it}}={\gamma m}_{i}\varepsilon_{t}+v_{\mathrm{it}}$, and $m_{i}$ is the ith row of the perturbation term weight matrix m, indicating that the spatial dependence of the green innovation activities is directly and negatively affected by the error perturbation items with a specific structure, then the SDM is degraded to a spatial error model (SEM).

## 4. Efficiency Evaluation of Green Innovation in Resource-Based Cities

### 4.1. Selection of Indicators of Green Innovation Efficiency

In constructing the input-output index system for the green innovation efficiency evaluation of resource-based cities, we draw on previous research results to fully consider the environmental factors and innovation factors, while taking into account the scientificity of the relevant indicators and the availability of data for resource-based cities. In terms of the input indicators, previous studies generally consider that the input of an activity factor mainly covers the three levels of the labor input, capital input, and energy input [25,26]. Among them, for the labor input, borrowing from Bai Xuejie et al. (2014), the number of personnel engaged in scientific and technological activities among unit employees is chosen as the scientific and technological labor input, and the number of personnel engaged in water conservation, the urban environment, and public service is the green labor input [27]. The capital input was expressed using the fixed asset investment. Since the total consumption of coal, oil, and natural gas in each city is not effectively counted, this paper draws on Li Yanjun (2014) and uses the electricity consumption of the whole society to measure the energy input [28]. In terms of the desired outputs, since urban green innovation is complex and involves numerous fields, including the three major systems of the economy, innovation, and the resource environment [29], it is important to consider the representative factors of the different aspects when comprehensively evaluating the output effects of green innovation in resource-based cities. In terms of the expected output, the GDP per capita of resource-based cities is used as the economic output indicator, the number of patents granted in cities is used as the innovation output indicator, and for the green output indicator, drawing on the ideas of Sun Yu (2015), the green coverage area of built-up areas is chosen as the measure of the green output capacity [30]. In addition, in terms of unintended outputs, urban wastewater emissions and SO_2_ emissions are used as the measures of the unintended outputs, according to the practice of Li Jian et al. (2019) [25]. The detailed indicator system is shown in Table 2.

### 4.2. Overall Evaluation of Green Innovation Efficiency

In this paper, the super-efficient SBM model considering the non-desired output is used to measure and obtain the mean value of the green innovation efficiency in resource-based cities from 2010–2019, as shown in Table 3.

Longitudinally, it can be seen from Table 3 that among the 64 resource-based cities studied in this paper, only nine cities, namely Laiwu, Huzhou, Daqing, Yichun, Qitaihe, Xinyu, Hegang, Liaoyuan, and Tongling, have average values of the green innovation efficiency on the production frontier during 2010–2019, while the remaining 55 resource-based cities are in the invalid state and the invalid state. These cities account for 85.94% of the study sample, indicating that the overall level of green innovation in resource-based cities is low. The 10 cities with the lowest efficiency are Linyi, Tonghua, Pingdingshan, Xingtai, Tangshan, Liupanshui, Handan, Zhangjiakou, Baise, and Bijie, and by observing the average value of efficiency, it can be seen that there is a more obvious variability among the green innovation efficiency of resource-based cities. Cross-sectionally, it can be seen from Table 4 that the average green innovation efficiency of resource-based cities from 2010–2019 shows a trend of fluctuation, followed by a slow increase. The year 2014 is the turning point of the trend of green innovation efficiency changes in resource-based cities during the study period and the second lowest point of the efficiency mean, with a difference of 0.2668 from the highest year of the efficiency mean in 2019. In November 2013, the State Council issued the Notice of National Sustainable Development Plan for Resource-based Cities (2013–2020), which requires each resource-based city to promote green development, strengthen environmental management and ecological protection, and support and lead the development of urban innovation and transformation. Since indicators such as patents have a lagging effect, the year following the issuance of the Plan, i.e., 2014, was in a transitional period for resource-based cities, and after 2014, the green innovation efficiency of resource-based cities slowly increased, indicating that the issuance of the Plan has made the green innovation transformation of resource-based cities effective, but the average value of the green innovation efficiency only reached 0.7296 in the highest year, which still has room for improvement at 27.04%.

### 4.3. Classification and Evaluation of Green Innovation Efficiency

To further analyze the differences in the green innovation efficiency of different resource-based cities, this paper divides each resource-based city into different groups according to the comprehensive planning classification and calculates the mean value of the green innovation efficiency of resource-based cities within each group, respectively, and the results are reflected in Table 5.

The sample classification of 64 resource-based cities in this paper covers four comprehensive classification types of resource-based city planning: the growth type, mature type, decline type, and regeneration type. As can be seen from Table 5, according to the comprehensive classification of planning, the green innovation efficiency of resource-based cities is the decline type, mature type, regeneration type, and growth type in descending order, among which the mean value of the green innovation efficiency of declining resource-based cities is always higher than the other three categories in each year from 2010 to 2019. However, on the whole, the green innovation efficiency of the four types of resource-based cities is at a low level, and only the average efficiency of declining resource-based cities and mature resource-based cities reaches above 0.5.

In addition, from the time series, the green innovation efficiency value of the four types of resource-based cities in the integrated planning classification has a clear upward trend in 2014, and the declining resource cities reached the highest point of the average efficiency at 0.9779 in 2019, only 2.21% away from the production frontier. Mature resource cities reached the highest efficiency value of 0.6791 in 2017, but there is still much room for improvement from the effective state. Regenerative resource cities also maintained a relatively stable growth in 2014. Growing resource cities reached a growth of 123.01% in 2015–2016, which, as the analysis shows, is caused by the fact that Songyuan City has greatly reduced the discharge of their industrial wastewater and increased their green output after 2015.

The Plan calls for categorizing and guiding the scientific development of various types of cities, including supporting the transformation and development of declining cities, guiding the innovative development of regenerating cities, promoting the leapfrog development of mature cities, and regulating the orderly development of growing cities. From this, it can be seen that the planning objectives are not the same for the development of different types of resource-based cities in 2013. Due to the gradual depletion of resources, declining cities are in urgent need of transformation and development, while regenerating cities are the focus of the transformation of resource-based cities, both of which need to be driven and led by innovation and pay attention to ecological and environmental issues in the process of economic development. In 2008–2009, the State Council announced 44 resource-depleted cities to help them achieve economic transformation and development, and the transformation of declining resource cities was carried out earlier and achieved certain results, forming a good foundation that made their green innovation efficiency from 2010–2019 higher than that of other types of cities. The green innovation efficiency of each type of resource city measured by the comprehensive classification of the Plan indicates that the declining resource cities currently meet the development requirements of the Plan in terms of green innovation, while the regenerative resource cities face greater difficulties and need to develop suitable strategies to enhance their green innovation level.

## 5. Analysis of the Influencing Factors of Green Innovation Efficiency in Resource-Based Cities

### 5.1. Selection of Indicators of Influencing Factors

(1)Environmental regulation intensity(er). Traditional management thinking suggests that environmental pollution controls raise the cost of controlling emissions and reduce the quality and competitiveness of a firm’s products. The Porter hypothesis states that tough and rational environmental policies can partially or fully offset these effects by encouraging firms to innovate, which in the long run helps to achieve a win-win situation of environmental improvement and competitive innovation. Therefore, the impact of environmental regulation on the efficiency of green technology innovation is due to a squeezing effect on technology innovation due to the increase in a firms’ production costs and an incentive effect due to the incentive to innovate technology to reduce costs. Referring to the practice of Cao Xia et al. (2015), this study selects “three waste utilization rates” as a proxy variable [31]. According to the availability of the pollution treatment data of prefecture-level municipalities, this paper uses three kinds of data, namely, the sewage treatment rate, comprehensive solid utilization rate, and household waste harmless treatment rate, and integrates them by the entropy value method to derive the characterization variable of the “three waste” utilization rate.(2)The level of government support (tech). The government funding support policy specifies the quantity and quality of green technology innovation, which in turn promotes the development of green innovation [31]. Technology investment as a fiscal regulation policy is a frequently used tool by the government to create a favorable technological innovation environment for enterprises and improve the urban technological innovation system on the one hand; on the other hand, it also has a reasonable and targeted direct impact on the total factor productivity of enterprises [25]. Although reasonable investment in science and technology can provide financial subsidies to enterprises to enhance their enthusiasm for technological innovation, some scholars believe that financial subsidies will squeeze enterprises’ R&D investment and make the conversion rate of technological innovation decline [20]. This study chose to select the proportion of government investment in science and technology to fiscal expenditure as the main proxy for the level of government support.(3)The level of industrial structure (ind). The development of a secondary industry has a great role in promoting the growth of the national economy, but it brings a high energy consumption industry and a lot of environmental pollution at the same time, such as steel, crude oil chemical products, metal machinery processing, and other excess product industries. While the enhancement of green innovation depends more on the reduction in environmental resources, the development of the tertiary industry with a low input, resource recycling, and a high pollution reduction, which drives the development of economic efficiency with the enhancement of the manufacturing structure, also reduces pollutant emissions and energy costs, thus promoting the enhancement of the green innovation efficiency. In this study, the proportion of the regional tertiary industry gross industrial output value to GDP is selected as a proxy for the level of industrial structure.(4)The level of external openness (fdi). The “pollution paradise theory” believes that pollution-intensive industries in other developed countries prefer to set up companies in developing countries or places with correspondingly lower technical standards for environmental protection, which is beneficial to local employment solutions but not to the optimization of the industrial structure; while the “double gap” theory states that regions with a higher level of external opening can be the first to master leading foreign technologies. Therefore, the technology spillover effect and pollution diffusion effect of opening up to the outside world cause opposite effects on the green innovation of the whole city. In this study, the ratio of the amount of actual foreign investment utilized in each region to the GDP of that region is chosen as a proxy variable for the level of foreign openness.(5)The level of economic development (dev). On the one hand, economic development brings financial security to the development of science and technology innovation, the improvement of the environmental resource structure, and the improvement of the urban environmental efficiency; on the other hand, since the level of economic and social development often reflects the highest standard of living of the inhabitants of a region, and the growth of average social income contributes to the preference for a better state of life, the choice of the inhabitants is also the main external driver of green efficiency improvement. Where the market economy is more mature, people are more sensitive to the environmental pollution emissions implied in traditional products and therefore turn to invest in environmental protection products [9,32], but this market choice is not feasible in all cases, and because green innovation products have longer life cycles, larger investments and higher risks, and higher market demand risks, it is often still easier for companies whose primary criteria are profit to see investment returns [33,34]. The study selects the relationship between regional gross domestic product (GDP) and the total number of companies. In this study, the ratio of gross regional product to the total regional population is selected as a proxy for the level of regional economic development. Specific influencing factor indicators are selected as shown in Table 6.

### 5.2. Spatial Autocorrelation Analysis Based on Moran’s I Index

First, based on GeoDa software to measure the Moran’s I index of 64 resource-based cities from 2010–2019, the geographic matrix was selected to test the spatial correlation of the green innovation efficiency in resource-based cities, and the cities that were not adjacent to the map were linked to the nearest resource-based cities to further correct the adjacency weight matrix, and the results are shown in Table 7.

In Table 7, it can be found that the values of Moran’s I index for the green innovation efficiency in resource-based cities from 2010–2019 are all positive and have passed the significance test at the maximum level, no higher than 10%, except for 2010 and 2012, which indicates that the green innovation efficiency of resource-based cities is not random, but there is a spatial dependence and corresponding spatial spillover effect, which is that the green innovation efficiency is more aggregated in geographically close cities and the levels of the green innovation efficiency are more similar. In addition, if we consider chronologically, the Moran’s I index value increased significantly in 2014 and kept rising continuously after 2014, which may be caused by the fact that after the introduction of the Plan in the year 2013, the state has unified the regulation of resource-based cities and the neighboring resource-based cities reached a strategic cooperation and coordinated development driven by the policy.

### 5.3. Analysis of Influence Factors Based on the Spatial Durbin Model

The statistical results of Moran’s I index show that there is a spatial correlation between the green innovation efficiency, indicating that it is feasible to study the factors influencing the green innovation efficiency through a spatial econometric model. Econometric tests were conducted for the sample of 64 resource-based cities as a whole and their four categories of cities were classified according to the different types, and the results are shown in Table 8. Firstly, the choice of a fixed effect and random effect is tested, and it can be found that the Hausman test rejects the original hypothesis of the random effect with a 1% significance level, so it is more appropriate to use the fixed effect. Secondly, for the fixed effect model, there also exists a corresponding three forms in spatial measurement: time fixed, space fixed, and time–space double fixed. The test result shows that the maximum likelihood function value (L: the higher the likelihood, the better the fit) means the time–space dual fixed-effects model is better than the time-fixed model and the space-fixed model). Finally, the Wald test is used to determine whether the spatial Durbin model (SDM) may degenerate into the spatial error (SEM) and spatial lag (SAR) models, and the results pass the 5% and 1% significance level tests, meaning that the SDM model cannot degenerate into the SEM or SAR model, i.e., the spatial Durbin model is identified as the best choice model. In summary, this paper selects the SDM model with dual fixed effects in time and space to test and analyze the influencing factors of the green innovation efficiency in resource-based cities.

As shown in Table 8, the spatial lag coefficient ρ of the green innovation efficiency of resource-based cities is 0.280, and it is significant at the 5% level, which indicates that for every 1% increase in the green innovation efficiency of the neighboring cities of resource-based cities, the green innovation efficiency of the city itself will increase by 0.280%, i.e., there is a positive spillover effect of the green innovation of neighboring cities on the city. In the classification of resource-based cities, the spatial spillover of growing resource cities is more obvious, probably due to the higher degree of cooperation reached among growing resource cities.

(1)There is a significant negative correlation between the intensity of the environmental regulations and the green innovation efficiency. In general, the standardized regression coefficient of the environmental regulation intensity measured by the “three waste” treatment rate is −0.016 and it passes the 1% significance test, indicating that the environmental regulation inhibits the improvement of the green innovation efficiency in resource-based cities. In terms of the classification of resource-based cities, the four categories of resource-based cities, namely mature, declining, regenerating, and growing cities, all have a significant negative relationship with the green innovation efficiency, with mature resource-based cities having the highest degree of influence. It can be seen that under the influence of government-led environmental protection policies, the production cost of enterprises subsequently increases, and its squeezing effect on technological innovation will make the green innovation efficiency of resource-based cities negatively affected. Therefore, for the use of environmental regulations in resource-based cities, a reasonable and bottom-up measure should be preferred to carry out, and only when environmental regulations have a positive incentive effect on technological innovation, if such a system is coordinated and effective.(2)There is a significant positive relationship between the level of government support and the green innovation efficiency. The overall regression coefficient of government support on the green innovation efficiency is 0.027 and it passes the 1% significance test, that is, for every 1% increase in government support, the green innovation efficiency of resource-based cities will increase by 0.027%. In terms of classification, mature, declining, regenerative, and growing resource-based cities all show a significant positive relationship with the green innovation efficiency, among which declining resource-based cities have the highest influence. The degree of green innovation is the highest. The government reduces the technological innovation risk of enterprises through direct financial support and other means and makes resource-based cities form a good innovation atmosphere to a certain extent, which motivates enterprises to carry out innovation activities and weakens the cost squeeze effect of green innovation.(3)There is a significant positive correlation between the level of industrial structure and the green innovation efficiency. Overall, the value of the regression coefficient of the industrial structure level expressed as the proportion of the regional gross industrial output value of the tertiary industry to GDP is 0.139 and it passes the 1% significance test, that is, for every 1% increase in the industrial structure level, the green innovation efficiency of resource-based cities will increase by 0.139%. From the classification point of view, mature resource-based cities, declining cities, regenerating cities, and growing cities have a significant negative relationship with green innovation, with mature resource-based cities having the highest degree of influence. This indicates that the more the industrial structure of resource-based cities tends to be reasonable, the more it is conducive to the improvement of ecological environmental protection and the technological innovation efficiency level, and the development of the tertiary industry for resource-based cities reduces both pollutant emissions and the pressure of resource extraction, so it is undoubtedly correct for resource-based cities to pay attention to the development of the tertiary industry for the sustainable development of resource-based cities.(4)There is a significant negative correlation between the level of foreign openness and the green innovation efficiency. Overall, the regression coefficient value of the level of foreign openness on the green innovation efficiency is −0.101 and it passes the 1% significance test, which means that enhancing the level of foreign investment utilization and expanding the openness of cities will, to a certain extent, reduce the green innovation efficiency of resource-based cities instead. In terms of classification, mature, declining, regenerating, and growing resource-based cities are all significantly and negatively correlated with the green innovation efficiency, with regenerating resource cities having the highest degree of influence. Foreign investment in resource cities has a squeezing effect on the local market, which not only squeezes the investment space of local industries but also leads to the phenomenon of “pollution paradise”, where more pollution-intensive enterprises settle in resource cities and cause pollution diffusion, which harms the local ecological environment.(5)There is a significant positive relationship between the level of urban economic development and the green innovation efficiency. On the whole, the regression coefficient value of the city’s economic development level is 0.082, and it passes the 1% significance test. A 1% increase in the city’s economic development level will increase the green innovation efficiency by about 0.082%; in terms of classification, mature, declining, regenerative, and growing resource-based cities all have a significant negative relationship with the green innovation efficiency, among which renewable resource-based cities have the highest degree of influence. Having a higher level of economic development in resource-based cities will not only bring financial security to local scientific and technological innovation development, environmental resource structure improvement, and upgrading, but also transform residents’ living needs, making them more inclined to choose green products and improve their requirements for a living environment, both of which force local enterprises to choose green innovation and thus improve the city’s green innovation level.

## 6. Conclusions

In this paper, using the panel data of 64 resource-based cities in China from 2010–2019 as a sample, we measured the green innovation efficiency of each resource-based city using the super-efficient SBM model with a non-expected output, and then conducted a classification study on the efficiency measurement results, and tested the spatial autocorrelation of resource-based cities using the global Moran’s I index, based on which the spatial Durbin model was used to test and analyze the factors influencing the green innovation efficiency of resource-based cities. The following conclusions were obtained.

(1)The evaluation of green innovation efficiency in resource-based cities. As a whole, the mean value of the green innovation efficiency of resource-based cities fluctuates and decreases from 2010 to 2014, and from 2014 to 2019, it shows a gentle upward trend, where 2014 is the turning point of the change in the mean value of the efficiency. Only nine cities, such as Laiwu City, among resource-based cities are at the production frontier surface, while all other cities have problems such as redundant factor inputs, an insufficient output efficiency, and the excessive output of negative environmental externalities during the sample period, and there is also a more obvious variability among the green innovation efficiency of each resource-based city. From the results of the analysis of the difference in the planning comprehensive classification, from 2010 to 2019, the four types of resource-based cities are at a low level from 2010 to 2019, and their green innovation efficiency is the decline type, mature type, regeneration type, and growth type in the order from high to low, among which the green innovation efficiency of declining resource-based cities is always at the highest level among all categories.(2)The influencing factors of the green innovation efficiency in resource-based cities. From the green factors, there is a significant negative correlation between the intensity of the environmental regulation and the green innovation efficiency, indicating that the environmental regulation inhibits the improvement of the green innovation efficiency in resource-based cities, and how to design an effective environmental protection system and give full play to the incentive effect of environmental regulation is an important choice for government departments. From the innovation factors, there is a significant positive correlation between government support and the green innovation efficiency, and improving government support is an effective way to improve the green innovation efficiency in resource-based cities. In terms of the comprehensive factors, the level of economic development of cities has a significant role in promoting the green innovation efficiency, indicating that the economic and scientific strength brought by economic development provides a strong guarantee for the improvement of the green innovation efficiency in resource-based cities. There is a significant negative correlation between the level of external opening and the green innovation efficiency, indicating that most resource-based cities in China at the present stage have become the transfer place of foreign low-level industries, which harms local development. There is a significant positive correlation between the level of industrial structure and green innovation efficiency, so the optimization and upgrading of industrial structure is an effective way to improve the green innovation efficiency of resource-based cities.

## 7. Recommendations

(1)Develop differentiated development strategies. Based on the characteristics of resource-based cities, we will develop differentiated strategies suitable for green development in each region. For example, the mature, regenerative, declining, and growing resource-based cities, as defined in the Plan, are different in terms of their resource endowment and ecological environment, and the degree and direction of their green innovation are different and needs to be improved. For cities that are seriously lagging behind in the green innovation efficiency, including regenerative and growing resource cities, we should focus our attention and support on understanding and solving the obstacles to green development and innovation development in these areas, reasonably planning the industrial structure, improving the access mechanism for high energy consumption and high pollution enterprises, and encouraging and supporting the development of clean and sustainable industries.(2)Formation of a regional cooperation mechanism. Since green innovation has a spillover effect, the formation of a perfect regional cooperation and mutual assistance mechanism is necessary to enhance the level of green innovation and reduce the differences in green innovation in resource-based cities. To address the problem of resource mismatch, an organization with complementary resources and mutually beneficial policies can be established for coordination, and bilateral or multi-governmental cooperation can be actively carried out. Drawing on the development experience of cities with a high efficiency in green innovation, strengthening the flow of outstanding talents between regions, and enhance their green innovation capacity breaks the local authorities with administrative regions as boundaries, establishes a long-term mechanism for coordinated regional development, and gradually reduces the gap between cities.(3)Strengthen the government’s scientific and technological support and set the intensity of environmental regulations scientifically and reasonably. Some examples of this are: continuing to enhance the policy support capacity, formulating appropriate environmental regulation measures, releasing the innovation vitality of enterprises, improving the innovation carrier capacity and leading role of SMEs, encouraging SMEs to increase research investment and set up research centers, guiding SMEs to carry out basic and cutting-edge R&D, and investing more funds in absorbing human resources and improving the innovation manufacturing efficiency.(4)Set the market access threshold for polluting enterprises and optimize the level of industrial structure. The local government will introduce corresponding policy measures guided by green innovative economic development and fully support the development of green industries in the city, promote high-tech industrial enterprises and green and environmentally friendly product enterprises in resource-based cities, and establish a strict review system and supervision for the entry of polluting enterprises, to achieve a green and sustainable development path for the city.

## Figures and Tables

**Table 1 ijerph-19-13772-t001:** Regional classification of resource-based cities.

Mode	Type	City
By Planning Comprehensive Classification	Growth type	Songyuan, Hezhou, Liupanshui, Bijie
	Mature type	Benxi, Jilin, Heihe, Daqing, Jixi, Mudanjiang, Huzhou, Suzhou, Bozhou, Huainan, Chuzhou, Chizhou, Xuancheng, Ganzhou, Yichun, Dongying, Jining, Tai’an, Laiwu, Sanmenxia, Hebi, Pingdingshan, Ezhou, Yunfu, Baise, Anshun, Chengde, Handan, Zhangjiakou, Xingtai
	Decline type	Fuxin, Fushun, Liaoyuan, Baishan, Yichun, Hegang, Qitaihe, Huaibei, Tongling, Jingdezhen, Xinyu, Pingxiang, Zaozhuang, Jiaozuo, Puyang, Huangshi, Shaoguan, Shuangyashan
	Regeneration type	Anshan, Panjin, Tonghua, Xuzhou, Suqian, Maanshan, Zibo, Linyi, Luoyang, Nanyang, Tangshan, Huludao

**Table 2 ijerph-19-13772-t002:** Green innovation efficiency index system of resource-based cities.

Target Layer	Tier 1 Indicators	Secondary Indicators	Tertiary Indicators	Unit
Green Innovation Efficiency	Inputs	Capital Investment	Fixed Asset Investment	million yuan
		Energy input	Electricity consumption of the whole society	Billion kWh
		Labor input	Number of personnel engaged in scientific and technological activities (scientific and technological labor input)	10,000 people
			Number of people engaged in water, urban environment, public services (green labor input)	10,000 people
	Outputs	Economic output	GDP per capita	Yuan
		Innovation Output	Number of patents granted	Pieces
		Green Output	The green coverage area of the built-up area	hectares
		Non-desired outputs	Wastewater discharge	million tons
			SO_2_ Emissions	ton

**Table 3 ijerph-19-13772-t003:** Mean value of green innovation efficiency in resource-based cities.

Sort by	City	Efficiency Value	Sort by	City	Efficiency Value
1	Laiwu City	1.6089	33	Taian City	0.5146
1	Huzhou City	1.1641	34	Hezhou City	0.4808
1	Daqing City	1.1582	35	Maanshan City	0.4633
1	Yichun City	1.1309	36	Jilin City	0.4472
1	Qitaihe City	1.1077	37	Benxi City	0.4269
1	Xinyu City	1.1039	38	Shaoguan City	0.4109
1	Hegang City	1.0948	39	Heihe City	0.4095
1	Liaoyuan City	1.0063	40	Zibo City	0.3979
1	Tongling City	1.0061	41	Luoyang City	0.3810
1	Dongying City	0.9803	42	Jiaozuo City	0.3744
1	Huaibei City	0.9505	43	Huainan City	0.3669
12	Baishan City	0.8600	44	Jining City	0.3645
13	Suqian City	0.8594	45	Ganzhou City	0.3448
14	Ezhou City	0.8585	46	Bozhou City	0.3415
15	Jingdezhen City	0.8285	47	Huludao City	0.3293
16	Chizhou City	0.8181	48	Yunfu City	0.3265
17	Xuancheng City	0.8045	49	Huangshi City	0.3246
18	Mudanjiang City	0.7145	50	Anshan City	0.2969
19	Puyang City	0.7144	51	Sanmenxia	0.2877
20	Jixi City	0.7140	52	Chengde City	0.2725
21	Chuzhou City	0.6650	53	Suzhou City	0.2559
22	Panjin City	0.6613	54	Yichun City	0.2459
23	Fushun City	0.6485	55	Linyi City	0.2269
24	Fuxin City	0.6393	56	Tonghua City	0.2117
25	Hebi City	0.6374	57	Pingdingshan	0.1825
26	Shuangyashan	0.5998	58	Xingtai City	0.1614
27	Anshun City	0.5912	59	Tangshan City	0.1554
28	Zaozhuang City	0.5844	60	Liupanshui City	0.1453
29	Xuzhou City	0.5831	61	Handan City	0.1406
30	Pingxiang City	0.5751	62	Zhangjiakou City	0.1393
31	Songyuan City	0.5611	63	Baise City	0.1368
32	Nanyang City	0.5154	64	Bijie City	0.1007

**Table 4 ijerph-19-13772-t004:** Annual average values of green innovation efficiency in resource-based cities.

Year	2010	2011	2012	2013	2014	2015	2016	2017	2018	2019
Efficiency	0.5129	0.4555	0.5218	0.4772	0.4628	0.5082	0.5908	0.7214	0.7087	0.7296

**Table 5 ijerph-19-13772-t005:** Average values of green innovation efficiency by type of resource-based cities (2010–2019).

Category	2010	2011	2012	2013	2014	2015	2016	2017	2018	2019	Average Value
Mature type	0.4840	0.4443	0.4948	0.4734	0.4525	0.5025	0.5366	0.6791	0.6161	0.6767	0.5360
Regeneration type	0.3621	0.3363	0.3489	0.3137	0.2714	0.3284	0.4394	0.6353	0.6033	0.5958	0.4235
Decline type	0.6979	0.6251	0.7049	0.6660	0.6781	0.7086	0.8197	0.9012	0.9762	0.9779	0.7756
Growth type	0.3496	0.1344	0.4192	0.1470	0.1446	0.1893	0.4222	0.4875	0.5156	0.4102	0.3220

**Table 6 ijerph-19-13772-t006:** Selection of indicators of influencing factors.

Subjects of Influence	Influencing Factors	Variable Abbreviation	Metrics
Green Factor	Environmental regulation intensity	*er*	The Utilization rate of “three wastes”
Innovation Factors	Level of government support	*tech*	The ratio of science and technology expenditure to fiscal expenditure (%)
Combined factors	Industry Structure Level	*ind*	Regional tertiary industry gross industrial output value as a proportion of GDP
	Level of the external opening	*fdi*	The ratio of the actual amount of foreign capital utilized in the year to the regional GDP by region (%)
	Economic Development Level	*dev*	The ratio of gross regional product to total regional population (yuan/person)

**Table 7 ijerph-19-13772-t007:** Moran’s I index for resource-based cities.

**Year**	**2010**	**2011**	**2012**	**2013**	**2014**
Moran’s I	0.062	0.055 *	0.084	0.075 *	0.166 *
**Year**	**2015**	**2016**	**2017**	**2018**	**2019**
Moran’s I	0.165 **	0.171 ***	0.184 **	0.189 *	0.203 ***

Note: ***, **, and * indicate passing the hypothesis test at 1%, 5%, and 10% significance levels, respectively.

**Table 8 ijerph-19-13772-t008:** Empirical results of influence factors of resource-based cities.

Variables	Overall	Mature	Declining	Regenerative	Growth Type
ρ	0.104 **	0.083	0.791 *	0.111 *	0.148 **
lner	−0.016 ***	−0.049 ***	−0.002 **	−0.080 *	−0.009 ***
Intech	0.027 ***	0.1054 **	0.2392 ***	0.1201 **	0.1612 *
lnind	0.139 ***	0.0012 ***	0.0021 *	0.1827 **	0.1062 *
lnfdi	−0.101 **	0.0431 *	−0.0211 **	−0.0216 ***	0.0051 **
lndev	0.082 **	0.1206 *	0.0971 *	0.1300 ***	0.0897 **
W × lner	−1.442 **	−1.345 ***	−1.312 *	−2.325 **	−1.470 **
W × lntech	1.002 **	1.465 **	1.008 ***	1.929 **	1.032 **
W × lnind	2.789 ***	2.568 *	2.905 ***	2.289 ***	1.780 **
W × lnfdi	−1.892 **	−1.904 ***	−1.872 **	−1.265 ***	−1.997
W × lndev	3.562 **	2.500 ***	2.987 ***	2.652 **	3.000 *
R-squared	0.745	0.491	0.424	0.500	0.621
L-likelihood	728.282	621.530	272.493	259.624	391.469
Hausman test	27.123 ***	28.773 ***	26.236 ***	33.467 ***	32.085 ***
Wald spatial lag	15.927 **	14.308 **	19.155 **	19.76 ***	14.31 ***
Wald spatial error	24.089 ***	24.024 ***	26.756 **	19.003 ***	17.461 *

Note: ***, **, and * indicate passing the hypothesis test at 1%, 5%, and 10% significance levels, respectively.

## Data Availability

Data are contained within the article.

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
