# Peer review of "Efficiency Evaluation and Influencing Factors of Green Innovation in Chinese Resource-Based Cities: Based on SBM-Undesirable and Spatial Durbin Model"

_ijerph, 2022, doi:10.3390/ijerph192113772_

Round 1

Reviewer 1 Report

The paper attempts to examine “Efficiency Evaluation and Influencing Factors of Green Innovation in Chinese Resource-based Cities: Based on SBM-Undesirable and Spatial Durbin Model”. First of all, I would like to thank the editor to select me as a reviewer for this paper. After reviewing, I find that this paper is interesting. The paper run numerous techniques, especially discussing on SBM-Undesirable and Spatial Durbin. In my opinion, this research is suitable, and able to accept in some aspects.

For better contribution to the literature, I have some revisions that are good for enhancing the quality of the manuscript.

1.      The introduction does not show the novelty of the paper as well as the methodology, the main results in the analysis, the structure of the study. I think this study needs a lot of improvement in the section of Introduction.

2.      The Green innovation efficiency index needs to be supported by previous studies (Table 2).

3.      More discussions for the utilization rate of "three wastes”. What are “three wastes”?

4.      The paper does not explain why you don’t use other common techniques? Common techniques may be good for doing the robustness check.

5.      The section 5.1, some discussions should be consistent with recent previous studies. It is good for showing the significant contribution of this study.

6.      The research findings are not compared to other studies, and reaffirm the novelty of the paper.

Thank you

Reviewer 2 Report

Thank you for sending this to my attention. The paper is interesting and shed light on an interesting topic. However, the paper is very technical and I would encourage authors to motivate and problematize this better in the introduction. Why do we need green city evaluations? I think this is taken for granted in the article. 

I also like the final section after conclusions on competing measurements. However, I would urge the authors say following Huemann et al., (2022) there is a need to also problematize and reconceptualize green cities and innovations making cities more green in a richer way.  A richer conceptualization may also find other causes and consequences driving these efforts further. While some of these processes are known and certain some are definitely more uncertain and unknown. 

Good luck

Reference

Huemann, M., & Pesämaa, O. (2022). The first impression counts:: The essentials of writing a convincing introduction. International Journal of Project Management, In press.

Round 2

Reviewer 1 Report

Dear Sir

I feel satisfied with this version

Thank you